# Questioning the lasting effect of galvanic vestibular stimulation on postural control

Mujda Nooristani [1,2,3]*, Maxime Maheu [1,2], Marie-Soleil Houde[1], Benoit-Antoine Bacon[4], François Champoux[1,2,3]

**1** École d'orthophonie et d'audiologie, Université de Montréal, Montréal, Québec, Canada, **2** CIUSSS Centre-Sud-de-l'Île-de-Montréal, Montréal, Québec, Canada, **3** Centre de recherche de l'Institut Universitaire de Gériatrie de Montréal, Montréal, Québec, Canada, **4** Department of Psychology, Carleton University, Ottawa, Ontario, Canada

* mujda.nooristani@umontreal.ca

**Data Availability Statement:** All relevant data are within the manuscript and its Supporting Information files.

**Funding:** This work was supported by the Natural Sciences and Engineering Research Council of

## Abstract

Noisy galvanic vestibular stimulation (nGVS) has been shown to enhance postural stability during stimulation, and the enhancing effect has been observed to persist for several hours post-stimulation. However, these effects were observed without proper control (sham condition) and the possibility of experimental bias has not been ruled out. The lasting effect of nGVS on postural stability therefore remains in doubt. We investigated the lasting effect of nGVS on postural stability using a control (sham) condition to confirm or infirm the possibility of experimental bias. 28 participants received either nGVS or a sham stimulation. Static postural control was examined before stimulation, immediately after 30 minutes of nGVS and one-hour post-stimulation. Results showed a significant improvement of sway velocity (p<0.05) and path length (p<0.05) was observed following nGVS, as previously shown. A similar improvement of sway velocity (p<0.05) and path length (p<0.05) was observed in sham group and no significant difference was found between nGVS group and sham group (p>0.05), suggesting that the observed postural improvement in nGVS could be due to a learning effect. This finding suggests the presence of experimental bias in the nGVS effect on postural stability, and highlights the need to use a sham condition in the exploration of the nGVS effect so as to disentangle the direct effect of the electrical stimulation from a learning effect. Furthermore, numerous parameters and populations need to be tested in order to confirm or infirm the presence of a real long-lasting effect of nGVS on postural stability.

## Introduction

Galvanic vestibular stimulation (GVS) is a technique used to stimulate the vestibular end organs and nerve by applying a low electrical current through electrodes placed over the mastoids (for a review see [1]). Animal studies have shown that GVS can increase or decrease the firing rate of the vestibular nerve, depending on the polarity of the current, and that it can also modulate vestibular function such as detection of head movements [1]. This approach can modulate vestibular reflexes by increasing the excitability of some reflexes [2].

Canada (NSERC) [grant number RGPIN-2016-05211].

**Competing interests:** The authors have declared that no competing interests exist.

Lately there has been a growing interest in a novel waveform of GVS, known as noisy GVS (nGVS), that involves applying a band-limited noise current. This approach has induced an enhancement of postural control in young and older adults, as well as in patients with bilateral vestibular loss [3–6]. The enhancement of postural control was operationalized and measured as a reduction of "Center of Pressure (CoP) parameters, such as sway velocity and path length. Indeed, an increase in the value of sway velocity and path length is related to an increase risk of falls [7, 8], therefore a reduction of those parameters is considered as an improvement of postural control. The putative mechanism underlying this postural enhancement is stochastic resonance; adding an optimal level of noise into a nonlinear system can enhance the detection of subthreshold signals and the processing of information [9]. Whether the observed postural enhancement persists over time remains, however, a matter of debate.

To our knowledge, only one study has evaluated the lasting effect of nGVS [10]. Fujimoto et al. [6] have shown that 30 min of nGVS induced an amelioration of postural stability in healthy older adults and that the effect could last several hours post-stimulation. Their experiment, however, did not include a control condition, and it is therefore possible that the enhancement reported might be caused by a learning effect and/or a placebo effect.

Indeed, it has been previously reported that Center of Pressure (CoP) measures can be improved by simple repetition [11–13]. The present experiment aims at investigating the lasting effect of nGVS on postural stability with a control condition so as to eliminate the possibility of the effect being due to experimental bias like a learning effect.

## Material and methods

### Participants

28 healthy young adults were randomly assigned to the nGVS group (n = 14; mean age: 23.28 (±3.58) years old) or the control sham group (n = 14; mean age: 23.69(±3.11) years old). There were no significant difference between groups for age (F(1,27) = 0.330; p = 0.571), height (F (1,27) = 0.572; p = 0.457) and weight (F(1,27) = 0.1.015; p = 0.323).

Each participant underwent a complete peripheral vestibular assessment that included the evaluation of semi-circular canals using the video head impulse test (vHIT: Eyeseecam, Interacoustics, Denmark), evaluation of both saccules with the cervical vestibular evoked myogenic potential (cVEMP: Eclipse EP-25/VEMP Interacoustics, Denmark), and evaluation of both utricules using ocular vestibular evoked myogenic potential (oVEMP: Eclipse EP-25/VEMP Interacoustics, Denmark). The cVEMP and oVEMP were considered normal when a replicable waveform was present at 95 dB nHL when using 500 Hz tone burst. For the vHIT, a vestibulo-ocular reflex (VOR) gain at 0.8 or higher was considered normal [14]. All participants had normal vestibular functions.

Research approval was obtained the 19th February 2018 from the Institutional Review Board of the Faculty of Medicine at the Université de Montréal (Comité d'éthique de la recherche en santé; IRB number: 17-178-CERES-D), and informed written consent was provided by all participants.

### Procedure

Participants had to perform a static postural control assessment with their eyes closed, standing barefoot on a foam surface (AIB Balance Foam, AIB, USA) placed on a force plateform (Accusway, AMTI, USA). This procedure, with eyes closed on a foam surface was selected as it allows to specifically measure the influence of vestibular inputs on postural control [15]. Three runs of 60 seconds CoP sway measurement were performed at three different measurement time points i) prior to stimulation (baseline), ii) immediately after the end of stimulation (T0:

0h post-stimulation) and iii) one-hour post-stimulation (T1: 1h post-stimulation). CoP sway was measured at the end of stimulation to examine the immediate effect of nGVS on postural control, and one-hour post-stimulation to assess the sustained effect of nGVS on postural control. Furthermore, previous studies have demonstrated that measuring CoP sway for 60 seconds increases test-retest reliability [16]. The CoP parameters recorded were sway velocity and path length; they were analyzed using Balance Clinic software (AMTI, USA).

Noisy GVS was applied using DC-Stimulator Plus (NeuroConn GmbHm Germany). Electrodes of 35 cm$^2$ (5 x 7 cm) in saline-soaked sponges were placed bilaterally over the mastoids. The stimulation intensity was set to 1mA as previous studies demonstrated that 1 mA of stimulation intensity increased cortical excitability [17]. Inuikai et al. [3] have also recently demonstrated that nGVS intensity fixed at 1 mA induced an improvement of postural control in young adults. The white noise ranged between 0 to 640 Hz and stimulation was applied continuously for 30 minutes. All subjects were sitting during the stimulation period to reduce the influence of an ongoing activity during the stimulation on the effect of nGVS post-stimulation. Furthermore, considering the duration of nGVS stimulation, the seated position prevents a fatigue effect that could occur during a standing position or during other position. Participants in the sham condition underwent the same experimental procedure but no stimulation was applied. The sham stimulation consisted of the current being ramped up to 1 mA for 30 seconds and then ramped down. This was done to create the same tingling sensation that can be perceived only during the ramp up of nGVS simulation and to make sham trials undistinguishable for nGVS trials. Moreover, despite the absence of electrical stimulation following the ramp down, participants had to keep the electrodes over the mastoids during 30 minutes. Therefore, it was not possible for the participant to determine if they received the sham or the real stimulation.

### Statistical analyses

Normalized ratios (NR) of postural improvement were calculated using sway data (sway velocity and path length). The normalized ratios were calculated as follow:

$$NR(Baseline) = \frac{Parameter\ at\ baseline}{Parameter\ at\ baseline}$$

$$NR(T0) = \frac{Parameter\ at\ T0}{Parameter\ at\ baseline}$$

$$NR(T1) = \frac{Parameter\ at\ T1}{Parameter\ at\ baseline}$$

To assess if the experimental effect was different from the sham condition, two separate repeated-measure ANOVA 2 Groups (nGVS; Sham) X 3 moments (Baseline; T0; T1) were performed for sway velocity and path length. Furthermore, post-hoc analysis using Bonferroni correction (p = 0.025) within each group for sway velocity and path length was performed to assess any improvement at T0 and T1 compared to baseline.

### Results

The repeated measures ANOVA 2 groups (nGVS; Sham) X 3 moments (Baseline; T0; T1) revealed no significant group difference for sway velocity ($F_{(1,26)}$ = 0.152; p = 0.700) nor path length ($F_{(1,26)}$ = 0.335; p = 0.567). Moreover, no significant group X time interaction was measured for sway velocity ($F_{(2,52)}$ = 0.419; p = 0.660) nor path length ($F_{(2,52)}$ = 0.540;

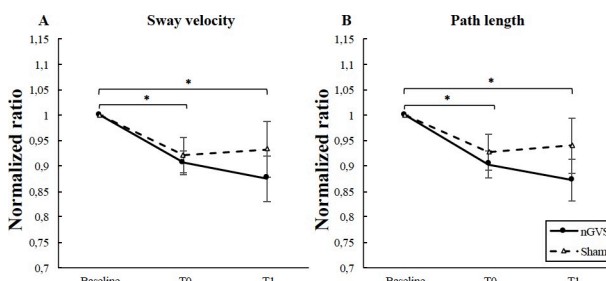

**Fig 1. (A) Sway velocity and (B) path length in nGVS group and sham group at each time point (Baseline; T0; T1).** Results suggest a significant improvement of sway velocity and path length in both groups at T0 and T1. Error bars represent the standard error of the mean. * = p<0.0001.

p = 0.586). However, a significant main effect of time was observed for sway velocity (F(2,52) = 5.918; p = 0.005) and for path length (F(2,52) = 5.789; p = 0.005).

A post hoc one sample t-test within each group revealed a similar improvement in both groups (Fig 1). The one sample t-test within the nGVS group revealed a significant difference between baseline and T0 for sway velocity (t(13) = 34.907; p<0.0001) and path length (t(13) = 34.823; p<0.0001). A significant difference was also observed between baseline and T1 for sway velocity (t(13) = 21.360; p<0.0001) and for path length (t(13) = 21.440; p<0.0001). The sample t-test within the sham group revealed a significant difference between baseline and T0 for sway velocity (t(13) = 26.784; p<0.0001) and path length (t(13) = 22.030; p<0.0001). A significant difference was also observed between baseline and T1 for sway velocity (t(13) = 21.360; p<0.0001) and for path length (t(13) = 22.480; p<0.0001).

## Discussion

The aim of the present study was to investigate the previously observed prolonged effect of nGVS on postural stability [10] with a control condition added so as to eliminate the possibility of the effect being due to experimental bias. As expected, our results revealed that postural stability following 30 minutes of nGVS resulted in a significant sustained improvement post-stimulation. This is in line with previous studies suggesting an improvement of sway performance after stimulation [10]. However, no significant difference between stimulation group and sham was observed at any time point, and a similar significant improvement post-stimulation was also observed in the sham group.

One possible explanation of the observed improvement of postural stability over time in the nGVS group could be a learning effect rather than an enhancing effect of nGVS. Previous studies have demonstrated a decrease in postural sway and in other sway parameters of the CoP with repeated testing of static postural control [12, 13]. Nordahl et al. [12] showed that a learning effect was observed in a population of normal healthy adults when the postural task was repeated multiple times. This learning effect was greater when subjects were standing on a foam rubber surface with their eyes closed and when the time interval between postural measures were short. An improvement of several CoP parameters was observed over time, particularly for path length and mean lateral and anteroposterior velocity. In the present study, the only postural sway condition tested was with the foam rubber surface and with eyes closed, which could therefore increase the possibility of a learning effect. Such a learning effect, however, might have been even greater in previous studies where postural control was assessed multiple times with different nGVS intensity to define the optimal intensity for each subject before the experimental protocol [5, 6, 10]. Such repetitions of the task before undergoing the experimental protocol could have maximized the presence of a learning effect. No matter the

procedure, the possible presence of such an important experimental bias underline the necessity of having a sham condition when examining the effect of nGVS.

Arguably, the absence of a significant difference between groups following nGVS could be due to a ceiling effect. All participants showed normal vestibular function, as assessed by clinical vestibular evaluations, and perhaps nGVS could not significantly enhance their performance. It is possible that nGVS might only have an enhancing effect on subjects with reduced vestibular function, as showed in previous studies [2, 4–6, 18].

An enhancing effect of 1 mA nGVS on postural sway in young adults has been previously demonstrated [3]. However, since the vestibular function of participants was not assessed in the aforementioned study, the effect of nGVS on individuals with normal vestibular function remains unconfirmed. It is highly possible that the effect of nGVS could have been more important in a population with a reduced vestibular function. Therefore, the examination of such a group, in comparison to a sham group, might also help to shed light into the lasting effect of nGVS on postural control by increasing the influence of the nGVS and eliminate the presence of a possible ceiling effect.

It could also be argued that the absence of improvement following nGVS might be related to the stimulation parameters applied, specifically *current intensity* and *current density*. Here, current intensity was fixed at 1 mA for all participants. This level has been found to induce a postural improvement in healthy young adults [3]. This procedure was also selected to avoid preliminary repetition before undergoing the experimental task, and to therefore reduce as much as possible the presence of a learning effect. In their experiment, however, Fujimoto et al. [6] adjusted the current intensity at the optimal level for each subject, in order to induce stochastic resonance to the peripheral vestibular system [7]. This procedure, however, can more easily generate a learning effect, as it requires several repetitions before proceeding with the experimental task. Another stimulation parameter to take into consideration is the current density applied, namely the amount of current flowing through the area stimulated which is related to the intensity of the current applied and to the size of electrode used. When investigating the effect of nGVS, these parameters are generally not reported (e.g. [2, 5, 6, 10]). Having no state-of-art method as a reference, we applied a low current density of 0.03 mA/cm$^2$. In retrospect, this might have had an impact on the results. Indeed, it has been demonstrated that current density is an important stimulation parameter when applying transcranial electrical stimulation and that higher levels of current density can have significant effect on corticospinal excitability [19]. A smaller electrode size might also be more effective in focusing on vestibular structures [20]. Taken together, one could argue that the use a current intensity greater than 1 mA or an electrode smaller than 35 cm$^2$ could have lead to a greater impact in the experimental group. This might need to be explored further to confirm the lasting effect of nGVS.

## Conclusion

In summary, the present study highlighted the necessity of incorporating a sham condition in the experimental design when investigating the effect of nGVS on postural stability, and more specifically with subjects presenting vestibular dysfunctions. The incorporation of a sham stimulation enables to dissociate the effect of stimulation from possible experimental bias. The lasting effect of nGVS remains to be confirmed with a sham condition, and with due consideration of ceiling effects and stimulation parameters.

## Supporting information

**S1 Dataset.**
(XLSX)

## Author Contributions

**Data curation:** François Champoux.

**Formal analysis:** Mujda Nooristani, Maxime Maheu.

**Funding acquisition:** François Champoux.

**Investigation:** Mujda Nooristani.

**Methodology:** Mujda Nooristani, Maxime Maheu.

**Supervision:** Maxime Maheu, François Champoux.

**Writing – original draft:** Mujda Nooristani, François Champoux.

**Writing – review & editing:** Maxime Maheu, Marie-Soleil Houde, Benoit-Antoine Bacon, François Champoux.

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
