## [Decision Letter · Decision Letter 0]

30 Sep 2019

PONE-D-19-25269

Questioning the lasting effect of galvanic vestibular stimulation on postural control

PLOS ONE

Dear Ms Nooristani,

Thank you for submitting your manuscript to PLOS ONE. After careful consideration, we feel that it has merit but does not fully meet PLOS ONE’s publication criteria as it currently stands. Therefore, we invite you to submit a revised version of the manuscript that addresses the points raised during the review process.

We would appreciate receiving your revised manuscript by Nov 14 2019 11:59PM. To enhance the reproducibility of your results, we recommend that if applicable you deposit your laboratory protocols in protocols.io, where a protocol can be assigned its own identifier (DOI) such that it can be cited independently in the future. For instructions see: http://journals.plos.org/plosone/s/submission-guidelines#loc-laboratory-protocols

We look forward to receiving your revised manuscript.

Kind regards,

Nicolás Pérez-Fernández

Academic Editor

PLOS ONE

**Journal Requirements:**

**Comments to the Author**

1. Is the manuscript technically sound, and do the data support the conclusions?

Reviewer #1: Yes

Reviewer #2: Yes

2. Has the statistical analysis been performed appropriately and rigorously? 

Reviewer #1: Yes

Reviewer #2: Yes

3. Have the authors made all data underlying the findings in their manuscript fully available?

Reviewer #1: No

Reviewer #2: Yes

4. Is the manuscript presented in an intelligible fashion and written in standard English?

Reviewer #1: Yes

Reviewer #2: Yes

5. Review Comments to the Author

Reviewer #1: This is an important and well performed study. This is a critical addition to the literature to reduce noise in the literature- as experiments must be performed with controls conditions or SHAM stimulation in brain stimulation studies.

Reviewer #2: It is a very nice and useful paper, which is also agreeable to read. However, soe corrections need to be implemented before publication. In particular the interpretation of the results needs to be improved.

Introduction

“Galvanic vestibular stimulation (GVS) is a technique used to stimulate the vestibular

system by applying an imperceptible level of electrical current through electrodes placed”

No, GVS can be perfectly perceptible, it depends of the intensity. Please correct.

Pease mention that GVS stimulate all vestibular sensors and the nerve. You should also quote adequate reviews on GVS in this first paragraph.

“postural enhancement” please be more precise

“might be caused by a learning effect.” Why do you exclude a placebo effect at that point?

Methods

“Research approval was obtained from the Institutional Review Board of the Faculty of

Medicine at the Université de Montréal, and informed written consent was provided by

all participants.” Pease give the IRB number, the exact reference of the ethical panel and the date of approval.

“Participants had to perform a static postural control assessment, standing on a foam

surface (AIB Balance Foam, AIB, USA) placed on a force plateform (Accusway, AMTI,

USA) with their eyes closed. Three runs of 60 seconds CoP sway measurement were

performed at three different measurement time points i) prior to stimulation (baseline), ii)

immediately after the end of stimulation (T0: 0h post-stimulation) and iii) one-hour poststimulation

(T1: 1h post-stimulation). The CoP parameters recorded were sway velocity

and path length; they were analyzed using Balance Clinic software (AMTI, USA).”

You never discussed in the introduction or here

• why you choose 60 seconds, using the foam,

• one hour of stimulation

• why the subjects were stimulated in a sited position and their posture tested in a standing position

Alternative choices were available. Also were the subjects barefoot?

“Therefore, it was not possible for the participant to determine if they received the sham or the real stimulation.”

How did you check that?

Results

No remark

Discussion

“As expected, our results revealed that postural stability following 30 minutes of nGVS resulted in a significant sustained improvement post-stimulation.”

Why is it certain that decreasing postural sway is an “improvement of stability”, at the very least in healthy subjects? It could be also an incidental effect in patients and why it is so sure that it is beneficial? Postural sway could be useful in many ways: exploring the limits of the polygon of sustentation, decreasing the risk that the plantar sol receptors become desensitized etc. Please delete that terminology, which is biased in the discussion and discuss these points. For instance, it could be some sort of freezing reaction, which explains why it was observed in the nGVS group.

In the conclusion, one should insist also of comparing the effect of GVS with sham procedures in vestibular patients.

6. PLOS authors have the option to publish the peer review history of their article (what does this mean?). If published, this will include your full peer review and any attached files.

Reviewer #1: No

Reviewer #2: No

---

## [Author Response · Author response to Decision Letter 0]

4 Oct 2019

We would like to thank you and the reviewers for the insightful comments that have been helpful towards significantly improving the manuscript. You will find below the detailed description of the changes we have made.

---

## [Decision Letter · Decision Letter 1]

10 Oct 2019

PONE-D-19-25269R1

Questioning the lasting effect of galvanic vestibular stimulation on postural control

PLOS ONE

Dear Ms Nooristani,

Thank you for submitting your manuscript to PLOS ONE. After careful consideration, we feel that it has merit but does not fully meet PLOS ONE’s publication criteria as it currently stands. Therefore, we invite you to submit a revised version of the manuscript that addresses the points raised during the review process.

We would appreciate receiving your revised manuscript by Nov 24 2019 11:59PM. To enhance the reproducibility of your results, we recommend that if applicable you deposit your laboratory protocols in protocols.io, where a protocol can be assigned its own identifier (DOI) such that it can be cited independently in the future. For instructions see: http://journals.plos.org/plosone/s/submission-guidelines#loc-laboratory-protocols

We look forward to receiving your revised manuscript.

Kind regards,

Nicolás Pérez-Fernández

Academic Editor

PLOS ONE

Additional Editor Comments (if provided):

Still comments 8 and 11 from Reviewer 2 must appear on the final text.

Reviewers' comments:

Reviewer's Responses to Questions

**Comments to the Author**

1. If the authors have adequately addressed your comments raised in a previous round of review and you feel that this manuscript is now acceptable for publication, you may indicate that here to bypass the “Comments to the Author” section, enter your conflict of interest statement in the “Confidential to Editor” section, and submit your "Accept" recommendation.

Reviewer #2: All comments have been addressed

2. Is the manuscript technically sound, and do the data support the conclusions?

Reviewer #2: Yes

3. Has the statistical analysis been performed appropriately and rigorously? 

Reviewer #2: Yes

4. Have the authors made all data underlying the findings in their manuscript fully available?

Reviewer #2: Yes

5. Is the manuscript presented in an intelligible fashion and written in standard English?

Reviewer #2: Yes

6. Review Comments to the Author

Reviewer #2: Thank you to have answered to my question. Please explain in the text and not only to me

- Why the subjects were stimulated in a seated position and their posture tested in a standing position?

- Why is it certain that decreasing postural sway in an “improvement of

stability”, at the very least in healthy subjects? It

7. PLOS authors have the option to publish the peer review history of their article (what does this mean?). If published, this will include your full peer review and any attached files.

Reviewer #2: No

---

## [Author Response · Author response to Decision Letter 1]

15 Oct 2019

We would like to thank the editor and the reviewers for the insightful comments that have been helpful towards significantly improving the manuscript. You will find in the filed "Response to Reviewer" the detailed description of the changes we have made. We have added Comment 8 and 11 in the text as suggested.

---

## [Editor Report · Decision Letter 2]

18 Oct 2019

Questioning the lasting effect of galvanic vestibular stimulation on postural control

PONE-D-19-25269R2

Dear Dr. Nooristani,

We are pleased to inform you that your manuscript has been judged scientifically suitable for publication and will be formally accepted for publication once it complies with all outstanding technical requirements.

With kind regards,

Nicolás Pérez-Fernández

Academic Editor

PLOS ONE

Additional Editor Comments (optional):

The authors have made a complete review now of the manuscript
---

## [Editor Report · Acceptance letter]

29 Oct 2019

PONE-D-19-25269R2 

Questioning the lasting effect of galvanic vestibular stimulation on postural control 

Dear Dr. Nooristani:

I am pleased to inform you that your manuscript has been deemed suitable for publication in PLOS ONE. Congratulations! Your manuscript is now with our production department. 

With kind regards,

on behalf of

Dr. Nicolás Pérez-Fernández 

Academic Editor

PLOS ONE